# Digital navigation of energy–structure–function maps for hydrogen-bonded porous molecular crystals

Chengxi Zhao[1,2], Linjiang Chen [2,3✉], Yu Che [2], Zhongfu Pang[2], Xiaofeng Wu[2,3], Yunxiang Lu[1], Honglai Liu[1], Graeme M. Day [4✉] & Andrew I. Cooper [2,3✉]

Energy–structure–function (ESF) maps can aid the targeted discovery of porous molecular crystals by predicting the stable crystalline arrangements along with their functions of interest. Here, we compute ESF maps for a series of rigid molecules that comprise either a triptycene or a spiro-biphenyl core, functionalized with six different hydrogen-bonding moieties. We show that the positioning of the hydrogen-bonding sites, as well as their number, has a profound influence on the shape of the resulting ESF maps, revealing promising structure–function spaces for future experiments. We also demonstrate a simple and general approach to representing and inspecting the high-dimensional data of an ESF map, enabling an efficient navigation of the ESF data to identify 'landmark' structures that are energetically favourable or functionally interesting. This is a step toward the automated analysis of ESF maps, an important goal for closed-loop, autonomous searches for molecular crystals with useful functions.

[1] Key Laboratory for Advanced Materials and School of Chemistry and Molecular Engineering, East China University of Science and Technology, Shanghai, China. [2] Leverhulme Research Centre for Functional Materials Design, Materials Innovation Factory and Department of Chemistry, University of Liverpool, Liverpool, UK. [3] Key Laboratory for Advanced Materials and Joint International Research Laboratory of Precision Chemistry and Molecular Engineering, Feringa Nobel Prize Scientist Joint Research Centre, School of Chemistry and Molecular Engineering, East China University of Science and Technology, Shanghai, China. [4] Computational Systems Chemistry, School of Chemistry, University of Southampton, Southampton, UK.
✉email: lchen@liverpool.ac.uk; g.m.day@soton.ac.uk; aicooper@liverpool.ac.uk

Hydrogen bonding is widely used for controlling supra-molecular assembly of organic building blocks[1,2] because it is directional and relatively strong for a non-covalent interaction. Molecules that combine hydrogen-bonding interactions and geometries that hinder close packing are known to promote porosity in crystalline molecular networks[3–6]. Indeed, there is a rapidly growing class of hydrogen-bonded organic frameworks (HOFs) with potential applications in gas storage and separation[7,8], molecular recognition[9,10], ion conduction[11,12], and catalysis[13].

Porous-bonded frameworks such as metal–organic frameworks (MOFs) and covalent organic frameworks (COFs) are assembled according to strong and predictable bonding patterns[14]. By contrast, porous molecular crystals are defined by the balance of many weak intermolecular interactions, such as hydrogen bonding and π–π stacking. As a result, small changes to the molecular structure can drastically change the crystalline packing of the molecule and its propensity for polymorphism, as well as the resultant physical properties. It is a long-standing challenge to control the crystallization of organic molecules to achieve specific structures with desired functions. The introduction of hydrogen-bonding groups, such as carboxylic acids, to create directional molecular building blocks or "tectons"[15] is one popular route for this, but such routes may also introduce synthetic complexity or chemical characteristics that are not aligned with the intended function (e.g., rigid, polar polyaromatic molecules can have very poor solubility). In the absence of a predictive understanding of molecular assembly in the solid state, it is challenging to rationally select or design appropriate molecular tectons for the synthesis of new functional molecular crystals—this is in sharp contrast to MOFs and COFs, for example, where intuitive iso-reticular design strategies have proved powerful[14].

Recently, we proposed the concept of energy–structure–function (ESF) maps to aid the discovery of porous molecular crystals with arresting properties[3]. To generate ESF maps, we combine crystal structure prediction (CSP), which determines the stable crystalline arrangements that are available to a molecule, with predictions of materials properties of interest. ESF maps, which are constructed using the molecular structure as the only input, reveal the possible structures and properties that are available for the molecule within the energetically accessible regions of its lattice–energy surface. This de novo strategy of exploring potential molecules using their predicted ESF maps is therefore applicable to both known and hypothetical molecules, and to any materials properties that can be computed from crystal structures such as gas adsorption and charge transport[16]. ESF maps can also be used to computationally pre-screen multiple candidate molecules for target applications to focus experimental efforts, which can often require months of synthetic work to access new molecular tectons. ESF maps have been shown to help guide synthetic control over pore size in isostructural porous organic cages[17–19] and to enable the discovery of new 'hidden' porous polymorphs of trimesic acid and adamantane-1,3,5,7-tetra-carboxylic acid, two archetypal molecules that had been studied for decades by crystal engineers[20]. The potential of small organic molecules to give rise to promising molecular photocatalysts[13] and electronics[16,21] may also be evaluated a priori by ESF maps.

Going forward, the fast yet accurate generation of ESF maps, as well as visualization and interpretation of the data, will require further development of techniques in fields that span computational chemistry, machine learning, and algorithms. First, the computational expense involved with CSP increases dramatically with the size and complexity of the molecule. For example, large, flexible molecules require extensive sampling of their coupled inter- and intra-molecular phase spaces in the search of stable crystal structures[13,22,23]. Second, materials properties that derive

from the crystal structure's electronic structure (e.g., band gap) or that require a long system equilibration (e.g., gas selectivity) can be very expensive to evaluate for large numbers of predicted structures, which is commonplace for the CSP landscapes of organic molecules. Third, it is challenging to explore the high-dimensional energetic, structural, and functional landscapes defined by an ESF map—in this respect, they differ from two-dimensional geographical maps.

Until now, ESF maps have usually been represented by projecting onto their corresponding CSP landscapes; that is, onto a plot of the crystal lattice energy as a function of the crystal density. This has proved powerful in highlighting functionally interesting structures that are also energetically favourable; for example, when there are pronounced local minima that are well separated from the bulk of the CSP landscape, sometimes referred to as "spikes"[3]. However, minima, or spikes, in the original high-dimensional ESF space could also be hidden in a simple one-dimensional representation, such as landscapes plotted against the crystal density or the pore surface area. One solution is to generate multiple ESF maps by 'cutting' through the ESF space along individual dimensions. Alternatively, more sophisticated structural representations—such as smooth overlap of atomic positions (SOAP) representations of atomic environments[24,25] and persistent homology barcodes of pore structures[26]—have been combined with machine learning techniques to learn two-dimensional representations of ESF maps.

Here, we explored the in silico computational design of a series of molecular tectons that comprise either a triptycene or a spiro-biphenyl core, functionalized with various different hydrogen-bonding moieties. Hydrogen bonding and π–π stacking were quantitatively analyzed for all the structures on the ESF maps to reveal how the maps evolve based on the different balance of intermolecular interactions in the various tectons. We show that the number of hydrogen bonding sites, as well as their position, has a profound influence on the resulting ESF maps. By applying unsupervised learning to pore descriptors, as well as SOAP representations, two-dimensional embeddings of the high-dimensional ESF data could be learned, which are human interpretable. ESF maps represented in this way enable the navigation of the complex ESF space within a unified framework, rather than using more traditional heuristics.

## Results

**CSP landscapes.** We studied a series of awkwardly shaped molecules with different hydrogen-bonding functionalities (Fig. 1, Supplementary Fig. 2). Following our previous study[3,27], we chose triptycene and spiro-biphenyl cores with the aim of frustrating close packing of the molecules in the solid state. To influence crystal packing, the molecular cores were functionalized by different hydrogen-bonding moieties. Benzimidazolones **T2** and **S2** are included here for comparison; **T2** was shown previously to afford stable, porous crystals. We also studied five six-membered-ring-based hydrogen-bonding moieties: 4-pyridone, 2-pyridone, 2,6(1H,3H)-pyridinedione, 2,4(1H,3H)-pyrimidinedione and 1,4(2H,3H)-pyrazinedione. In combination with the triptycene core, five new molecules were generated: quinolones **TH1** and **TH2**; isoquinolinedione **TH3**; quinazolinediones **TH4** and **TH5** (Fig. 1). For the spiro-biphenyl core, only two molecules were considered: quinolone **SH1** and quinazolinediones **SH2** (Supplementary Fig. 2). These seven molecules bear different numbers and ratios of hydrogen-bond donors and acceptors, offering a potentially diverse array of options for intermolecular hydrogen bonding and crystal packing.

**TH5** has been synthesized before[28], while **TH1–TH4**, **SH1** and **SH2** are, in theory, accessible experimentally via known organic

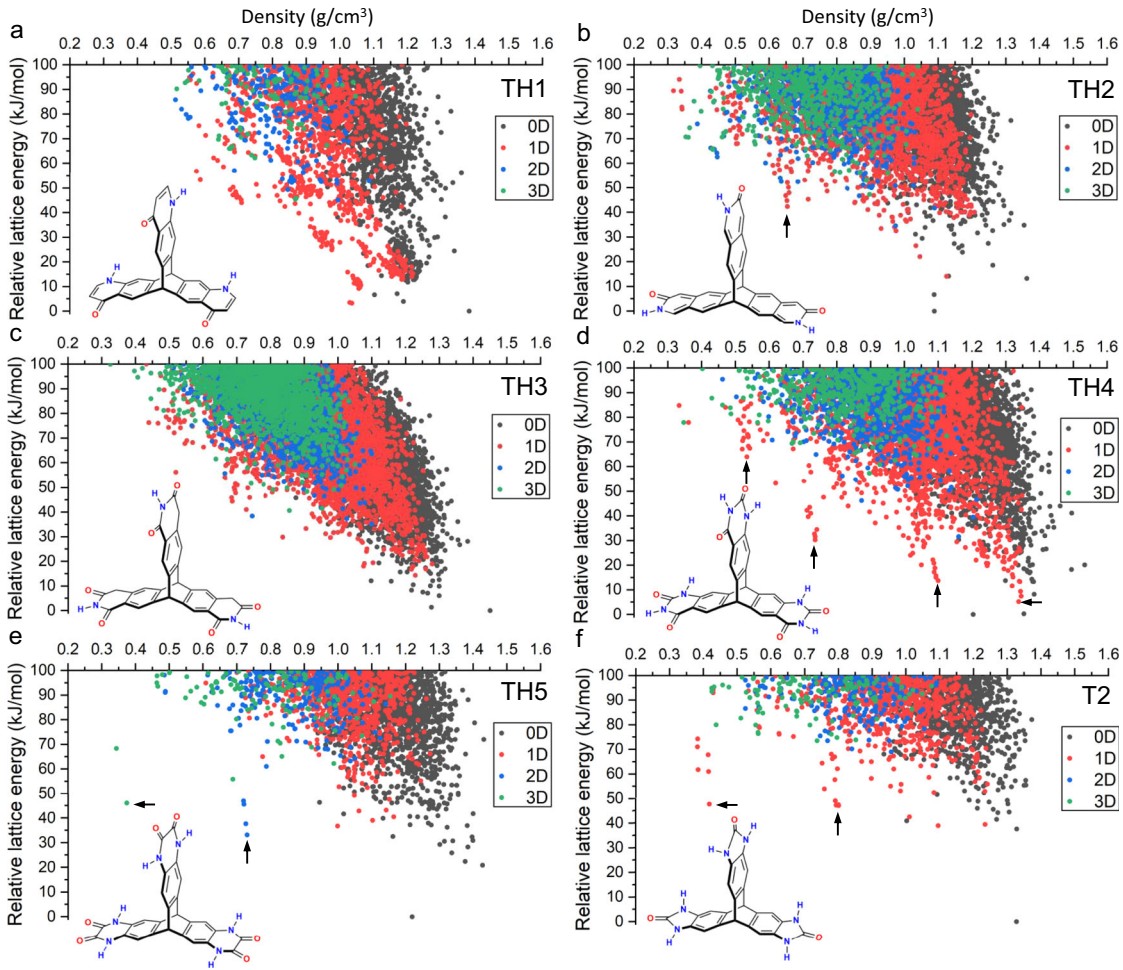

**Fig. 1 Energy–structure–function maps. a–f** Crystal structure prediction energy–density plots for the molecular building blocks shown in the figure: **TH1** (**a**), **TH2** (**b**), **TH3** (**c**), **TH4** (**d**), **TH5** (**e**) and **T2** (**f**). Each point corresponds to a computed crystal structure. The symbols are colour coded by the dimensionality of the pore channels, assessed using a probe radius of 1.7 Å; see Supplementary Fig. 1 for alternative plots with shuffled plotting orders for the points. Molecules **TH1–4** each have two isomers arising from the arrangement of the hydrogen-bonding moieties on the triptycene core; only the higher-symmetry isomers were considered here. Arrows indicate the spikes that are referred to in the text.

reactions (Supplementary Figs. 16–21). However, we envisage that some syntheses might be elaborate and challenging—for example, in terms of isolating specific isomers—and also because these rigid aromatic molecules often have poor solubility. As such, computational pre-screening prior to experiments has significant value. Molecules **TH1–TH5**, **SH1–SH2** may undergo keto–lactam to enol–lactim tautomerization via intra- or intermolecular proton transfer. In solution, the lactam–lactim equilibrium is dependent on the solvent polarity, which is shifted to lactam in polar solvents[29]. In the solid state, the lactam form is often found to dominate;[30] specifically, the molecular arms of **TH1–TH5** have been reported in their corresponding lactam form in the Cambridge Structural Database (deposition numbers: 643895, 787295, 1178376, 702449 and 1178443). We did not attempt any organic synthesis in this study, but we offer these systems and the associated predictions as experimental targets for the future.

Computational methods for CSP involve a global exploration of the multidimensional lattice energy surface for stable energy minima, followed by an assessment of the relative stabilities of the resulting structures. Here, unbiased searches of the lattice energy surface[31] were used to determine the stable crystalline arrangements that are available to each of the molecules (Fig. 1, Supplementary Fig. 3). Organic molecules tend to pack densely

to maximize their intermolecular interactions, reducing the energetic cost of void space in a solid. As such, generating porosity in molecular crystals remains a challenging task for crystal engineering. Having a rigid and contorted molecular shape may not always be sufficient to prevent dense packing. Indeed, most low-energy structures of triptycene are non-porous and the lower edge of the energy–density distribution decreases nearly monotonically, as is typical for most organic molecules[3]. The 'leading edge' of a CSP landscape comprises structures with the lowest energy at a given density, and stable porous structures have previously been realized experimentally in this region[3].

The CSP landscape for **T2** is markedly different to that of triptycene, with multiple low-density structures predicted to be substantially lower in energy than the bulk of the landscape, forming the so-called 'spikes' (Fig. 1f). The emergence of spikes from the bulk of a CSP landscape indicates that the molecule may form unusually stable crystalline structures for their respective densities, and the shape of the energy–density distribution suggests a large energetic barrier separating these structures from higher-density regions of the landscape. For **T2**, the minimum-energy structures within the two spikes at densities of about 0.4 g cm$^{-3}$ and 0.8 g cm$^{-3}$—**T2-γ** and **T2-β**, respectively—can be accessed experimentally by solvent stabilization, even though they are about 50 kJ mol$^{-1}$ above the global energy minimum[3]. Despite using a smaller selection of space

groups for CSP here than previously (we used 23 out of the 89 space groups used in ref. [4] see "Methods" section), the partial energy–density landscape of **T2** shown in Fig. 1f captures the same key features as the landscape sampled more exhaustively, including the major 'spikes' and the four experimental polymorphs (**T2-σ**, **T2-β**, **T2-γ**, and **T2-δ**). We therefore carried out CSP in these 23 space groups for all the other molecules in order to reduce computational costs.

The leading edge of the energy–density landscape of **TH1** decreases nearly monotonically, with no structures having a density below 0.5 g cm$^{-3}$ located within 100 kJ mol$^{-1}$ above the global energy minimum (Fig. 1a). **TH2** is a positional isomer of **TH1**: this arrangement of hydrogen-bonding sites broadens the density distribution of the predicted structure landscape to lower densities and a spike appears at around 0.65 g cm$^{-3}$ (Fig. 1b). The isoquinolinedione, **TH3**, has one extra carbonyl group per arm compared to **TH1** and **TH2**, and a methylene unit in the 6-membered ring. The addition of three additional hydrogen-bond accepting groups in **TH3** with respect to **TH2** does not seem to promote low-density, stable structures (Fig. 1c). By contrast, the energy–density distribution for **TH4** (Fig. 1d) is reminiscent of that for **T2** (Fig. 1f) and shows multiple low-energy spikes. Three spikes are apparent at densities of about 0.5 g cm$^{-3}$, 0.7 g cm$^{-3}$, and 1.1 g cm$^{-3}$, which are 63.9 kJ mol$^{-1}$, 30.0 kJ mol$^{-1}$, and 13.6 kJ mol$^{-1}$ above the global energy minimum, respectively. By analogy with **T2**, these structures fall in an energy range that we would expect might be accessible via solvent stabilization. **T2** does not have any predicted structures with one-dimensional (1D) channels (red points in Fig. 1) within 30 kJ mol$^{-1}$ above the global minimum (Fig. 1f; see also Fig. 2c in ref. [4]). By contrast, the plot for **TH4** shows a significant number of structures with 1D pore channels in the density range 1.25–1.35 g cm$^{-3}$; the minimum-energy structure among these is just 5.1 kJ mol$^{-1}$ above the global minimum. The spikes on the landscape of **TH4** can also be recognized at similar density regions on the landscape of **TH2**, although they are less pronounced. Among the four triptycene-based molecules, the positioning of the hydrogen-bonding groups (**TH2** vs. **TH1**) appears to play a more significant role in promoting porosity than their number (**TH2** vs. **TH3**).

**TH5** is a positional isomer of **TH4** and has a higher point symmetry of **D$_{3h}$** (c.f., **C$_{3v}$** for **TH4**). Two pronounced spikes emerge from the landscape at densities of about 0.35 and 0.7 g cm$^{-3}$ (Fig. 1e), with the minimum-energy structure in the lowest density spike being only 46.0 kJ mol$^{-1}$ above the global energy minimum. This energy gap is comparable to that (47.6 kJ mol$^{-1}$) for the lowest-density experimental polymorph of **T2**, **T2-γ** (minimum-energy structure in the spike at 0.5 g cm$^{-3}$), indicating the possibility of realizing this low-density structure of **TH5**. In contrast to **TH2**, **TH4** and **T2**, where the spikes mainly contain structures with 1D pore channels, structures in the spikes for **TH5** show higher (2D or 3D) pore connectivity (Fig. 1e).

The energy–density landscapes for **SH1** and **SH2** (Supplementary Fig. 3) show far fewer predicted structures within 100 kJ mol$^{-1}$ of the global energy minimum than their triptycene counterparts bearing the same hydrogen-bonding motifs (**TH2** and **TH4**, respectively). Likewise, **S2**, having the same hydrogen-bonding moieties as **T2**, does not show unusually stable low-density structures. This suggests that spiro-linked tetrahedral geometries are less effective at generating porosity.

**Hydrogen bonds stabilize porous structures**. Analysis of the intermolecular hydrogen bonding in the leading-edge **T2** structures revealed that structures within the spikes feature hydrogen bonded networks with 2D rings propagating along a third direction to form one-dimensional pore channels[3]. Here, we set out to perform quantitative analyses of the hydrogen bonding in the predicted structures of all the molecules studied here (Fig. 2a, c, e and Supplementary Fig. 4). A hydrogen bond is defined here for an interacting system of three atoms N–H•••O—where, the hydrogen atom (H) is covalently bonded to the nitrogen atom (N) and is interacting with the oxygen atom (O)—when the distance between H and O is shorter than the sum of their van der Waals radii minus 0.1 Å and the angle formed by N–H•••O, centred on H, is larger than 100°[3].

Figure 2a, c, e shows the CSP landscapes of **T2**, **TH4** and **TH5**, colour coded by the number of hydrogen bonds each molecule forms in the corresponding crystal structure; this number is (by definition) the same for all the molecules in a given crystal structure because only crystal structures with one symmetrically unique molecule were considered in these CSP calculations (Z′ = 1). The analogous results for the other molecules studied are shown in Supplementary Fig. 4. The number of hydrogen bonds for a molecule accounts for both cases when carbonyl groups act as a hydrogen-bond acceptor and when N–H groups act as a hydrogen-bond donor. For example, the maximum value of the number of hydrogen bonds for a single **T2** molecule is 12: that is, the six N–H groups can each participate in one hydrogen bond, while the three O atoms can each participate in two hydrogen bonds. In a similar way, we also quantified the extent of intermolecular stacking in each predicted crystal structure (Fig. 2b, d, f and Supplementary Fig. 5) by counting the π–π stacking modes formed between the arms of the various molecules. Here, we only consider co-facial and parallel-displaced stacking conformations but not T-shaped ones.

Across the whole series of molecules, intermolecular hydrogen-bonding and intermolecular stacking (see "Methods" section for the specific definitions used in this study) are found to be mostly competing or orthogonal forces in driving the solid-state packing of these molecules (Fig. 2 and Supplementary Figs. 4, 5): i.e., most structures—particularly in the bulk of the CSP landscape—do not simultaneously show a large number of hydrogen bonds and a large number of π–stacked molecular arms. This results from the positioning of the hydrogen-bonding motifs in the molecule, together with the contorted molecular core. However, this simple picture is more mixed for structures that are close to the leading edge of the landscape or within the spikes. For **T2** and **TH5**, such structures are primarily stabilized by extensive hydrogen bonding (Fig. 2a, e), except for some **T2** structures in the medium density range (around 0.8 g cm$^{-3}$) that show enhanced but still moderate stacking between the molecular arms (Fig. 2b). By contrast, the leading-edge structures of **TH4** benefit from both strong hydrogen bonding and moderate-to-strong molecular stacking (Fig. 2c, d), except for the lowest-density spike (<0.4 g cm$^{-3}$) where structures only exhibit strong hydrogen bonding. For all the molecules, densely packed structures in the bulk of the landscape are characterized by increased levels of intermolecular stacking and decreased levels of intermolecular hydrogen bonding. The conclusions for the spiro-linked **SH1**, **SH2** and **S2** molecules are broadly the same as for their triptycene analogues (Supplementary Figs. 4 and 5).

**ESF data mapped onto individual structural descriptors**. ESF maps combine CSP, which determines the stable crystalline arrangements available to a molecule, with predictions of materials properties of interest, using the molecular structure as the only input (see "Methods" section for details). Conventionally—and intuitively—ESF maps are projected on their corresponding CSP energy–density landscapes, with each point on the 'map' representing a predicted crystal structure with its colour coded to one of its physical or functional properties; for example, the pore

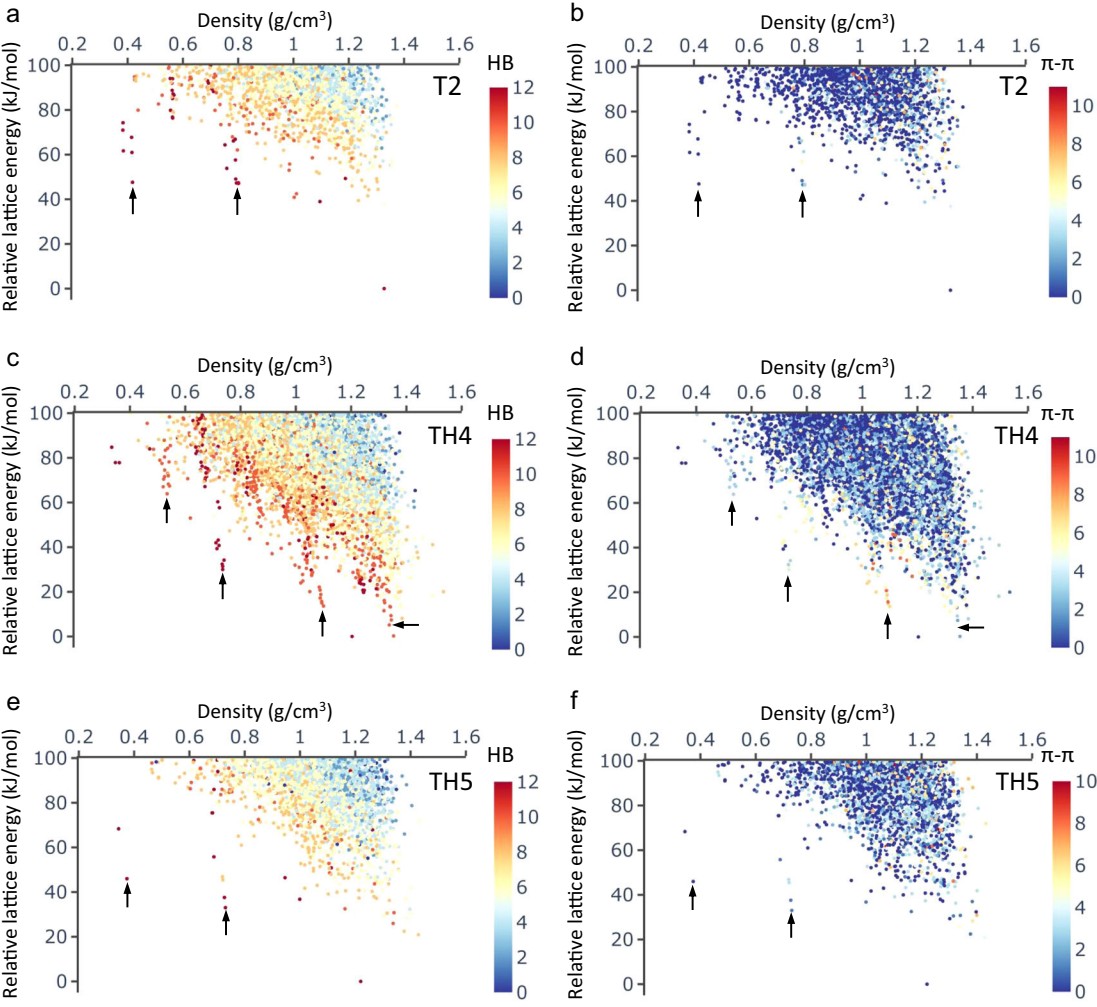

**Fig. 2 ESF maps for intermolecular hydrogen bonding and intermolecular stacking.** CSP energy–density landscapes, colour coded by the number of intermolecular hydrogen bonds (HB; **a**, **c** and **e**) or the number of intermolecular stacking modes ($\pi$–$\pi$; **b**, **d** and **f**; defined as face-to-face stacking between two molecular arms) formed by one molecule with its neighbours in the crystal structure: **T2** (**a**, **b**), **TH4** (**c**, **d**) and **TH5** (**e**, **f**). Arrows indicate the spikes that are referred to in the text.

topologies are colour coded in Fig. 1. This is not the only possible representation: more generally, an ESF map can be projected onto many different structural parameters. Figure 3 shows ESF maps projected onto three different structural descriptors: crystal density, largest free sphere diameter, and accessible surface area.

For **TH4**, spikes emerge from the bulk of the landscape on all three ESF maps, as shown in Fig. 3a–c. Low-energy structures within these spikes show complete or almost complete saturation of the hydrogen-bonding sites of the **TH4** molecule, showing that extensive intermolecular hydrogen bonding serves to facilitate stable porous structures. The minimum-energy structure of each pronounced spike in the energy–density landscape is shown in Fig. 3d; these structures are also found on the leading edge of the landscape when plotted against the largest free sphere diameter (Fig. 3b) or the accessible surface area (Fig. 3c). These landmark structures (Fig. 3d; A–F) all exhibit extended hydrogen-bonded chains along the pore channels. In **TH4-A**, molecules pack 'head-to-head' to form two-dimensional layers, using the hydrogen-bonding sites at the tip of each arm (Supplementary Fig. 6a); these layers stack along the third direction, forming linear hydrogen bonds between the edges of the molecules. Similar hydrogen-bonding patterns also appear in the other landmark structures (Fig. 3d), with stacking between the molecular arms becoming more extensive as the structure gets denser (Supplementary Fig. 6).

**TH5** is predicted to yield landmark structures **A**, **B**, **C1** and **D2** (Supplementary Fig. 8) that are isostructural with **TH2/4-A** to **D**, respectively, in terms of the 1D channel shapes. In contrast with **TH2/4-A** having 1D pore channels, the 1D channels in **TH5-A** are interconnected through apertures in the pore 'walls', as a result of the packing of **TH5** molecules along the channel direction (Fig. 4). Similarly, interconnected 1D channels are present in other **TH5** landmark structures, such as **B**, **C1**, **C2**, **D1** and **D2** (Supplementary Fig. 9). **TH5-A** has a predicted density of just 0.374 g cm$^{-3}$, with a calculated accessible surface area of 4447 m$^2$ g$^{-1}$, assessed by a probe radius of 1.70 Å. This highly porous structure might be accessible in the laboratory because it is isostructural to **T2-γ**, which has been isolated[3,27], and it is predicted to have a similar relative stability (46.0 and 47.6 kJ mol$^{-1}$ above the corresponding global minimum for **TH5-A** and **T2-γ**, respectively). If it can be prepared and it is stable to desolvation, **TH5-A** would be one of the lowest density molecular crystals reported to date. Few (if any) desolvated molecular crystals have densities lower than 0.4 g cm$^{-3}$. Two triptycene-based HOFs, reported by Stoddart and co-workers[32,33], showed ultra-low framework densities of 0.323 or 0.231 g cm$^{-3}$, but both of the solved crystal structures were for solvates. One of these crystals was reported to have a theoretical surface area of 1690 m$^2$ g$^{-1}$, although the measured Brunauer–Emmett–Teller

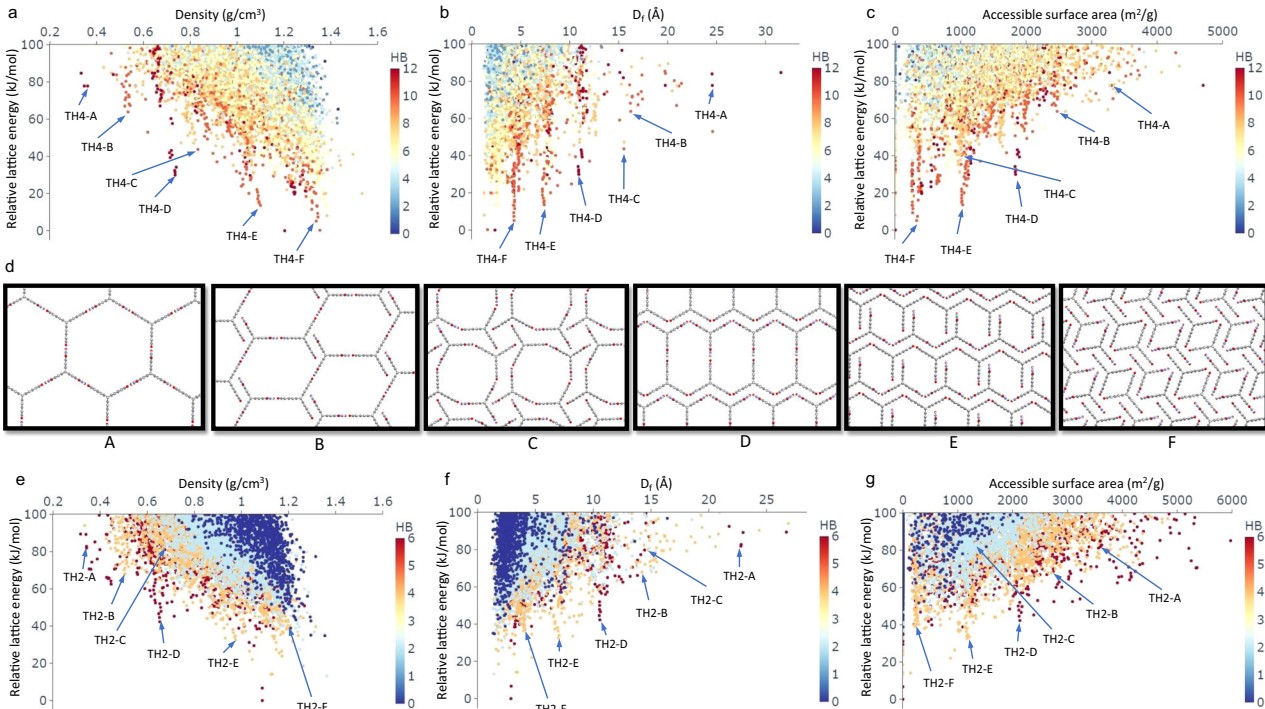

**Fig. 3 ESF maps for individual structural descriptors.** ESF maps for **TH4** (**a**–**c**) and **TH2** (**e**–**g**), plotted against the crystal density (**a**, **e**), the largest free sphere diameter ($D_f$; **b**, **f**) or the accessible surface area (**c**, **g**); symbols are colour coded by the number of hydrogen bonds formed by each molecule in the crystal structure. Selected **TH4** 'landmark' structures A–F are displayed in (**d**) and labelled in (**a**–**c**), with their **TH2** analogues labelled in (**e**–**g**).

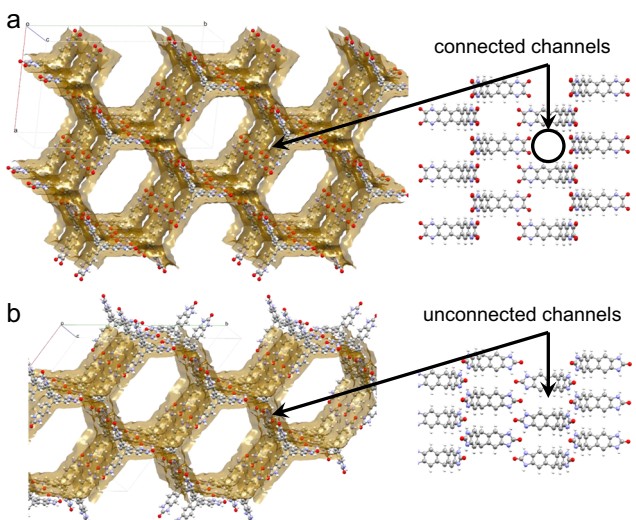

**Fig. 4 ESF maps predict a highly porous solid for TH5.** Solvent accessible surfaces (left) of **TH5-A** (**a**) and **T2-γ** (**b**). **TH5-A** shows a three-dimensionally interconnected pore space within the structure. Unlike for **T2-γ**, the 1D hexagonal pore channels in **TH5-A** are predicted to be connected by apertures in the pore walls that are orthogonal to the direction of the channels; one such aperture is indicated by the black circle on the right-hand-side figure. Predicted surface area for **TH5-A** = 4447 m² g⁻¹ (cf., 3199 m² g⁻¹ predicted for **T2-γ**).

surface areas were much lower. We therefore suggest that **TH5-A** has the potential to the most porous HOF to date, although its very low predicted density implies that careful desolvation might be required; for example, by using solvent exchange protocols or supercritical drying.

**TH2** gives similar ESF maps to those of **TH4**: spikes emerge from the landscape in the same regions of the structural descriptor used (Fig. 3e–g). This is because **TH2** is predicted to generate crystal structures **TH2-A** to **F** that are isostructural with **TH4-A–F**, respectively, in terms of the shapes of the one-dimensional pore channels (Fig. 3d and Supplementary Fig. 7); for example, **TH2-A** and **TH4-A** both have hexagonal pore channels. However, in these **TH2** landmark structures, molecules do not pack 'edge-to-edge', due to the absence of the hydrogen-bonding sites on the edges of the molecule. Instead, **TH2** molecules tend to form staggered hydrogen-bonded chains along the pore channels: each molecular arm forms hydrogen bonds with two other arms from two different molecules (Supplementary Fig. 7). This 'head-to-tail' hydrogen-bonding motif[34], labelled "type 2" in Supplementary Fig. 7g, h, helps the molecular assembly to extend by repeating the bonding motif. **TH2-A–F** are mostly found on the leading edge of the landscape plotted against one of the structural descriptors (Fig. 3e–g); **TH2-C** is higher in lattice energy than the corresponding region of the leading edge, for all three ESF maps. In line with the above discussion for **TH4**, **TH2** structures on the leading edge and within the spikes—particularly low-density, large-pore, or large-surface-area ones—exhibit rich intermolecular hydrogen bonding. All six hydrogen-bonding sites on each **TH2** molecule are used in **TH2-A**, **D**, while four hydrogen-bonding sites are used in **TH2-B**, **C**, **E**, **F**.

Decomposition of the lattice energy into its physical contributions (Supplementary Fig. 10) corroborates the picture built by simple counting of the intermolecular hydrogen bonds and π–π stacking modes. All landmark structures are characterized by strong, stabilizing electrostatic interactions, with the **TH4** structures consistently more stable than their **TH2** counterparts thanks to its larger number of hydrogen-bonding sites than **TH2**. Structures **A**, **B** and **D** bear (nearly) linear hydrogen bonds and hence are stabilized by strongly directional electrostatic interactions, while structures **E** and **F** show enhanced dispersion

interactions resulting from increased stacking between the molecular arms. The landmark structures of **TH2** and **TH4** are reminiscent of the experimental polymorphs of **T2**: structures **TH2/4-A, C, D** and **E** have isostructural pore channels with **T2-γ, α, β** and **δ**, respectively. Therefore, it is conceivable that these landmark structures—particularly for **TH4**, whose landmark structures are all minimum-energy structures within their corresponding spikes—might be experimentally accessible should this molecule be synthesized in the future.

To assist with both the analysis in this study and with future interpretations of ESF maps, we developed an interactive visualization tool—an ESF Explorer—using **TH4** as an example here (https://www.interactive-esf-maps.app). This tool allows the user to interrogate the correlations, dependencies and relationships between the various dimensions of the data. In the ESF Explorer, a variety of 'descriptors' can be chosen as the X-axis, the Y-axis and as colour-coding in the ESF map. The predicted crystal structures are displayed interactively when points are selected on the ESF plot. Our interactive visualization tool was inspired by the pioneering efforts of Moghadam et al. in exploring high-throughput screening data of MOFs[35,36].

**Two-dimensional embeddings of the high-dimensional ESF data.** While projecting an ESF map onto individual dimensions is a useful way of exploring data, it can be laborious when many structural and functional properties are associated with 1000 s to 10,000 s of structures typically on a single ESF map, even with the help of our interactive ESF Explorer. It is therefore desirable to devise a simple and general approach to represent the high-dimensional data of ESF maps, allowing us to systematically identify 'landmark' structures on the map, be they either energetically favourable or functionally interesting structures. To do this, we encoded each of the crystal structures on an ESF map by a number of pore descriptors including pore diameters, surface areas and some variants of these in order to capture, to some extent, the heterogeneity of pore/channel sizes within a given map (see Supplementary Methods). We then used the affinity propagation algorithm[37] to cluster all the crystal structures into unique groups on the porosity space defined by these pore descriptors. For each group, a landmark structure was identified as the lowest-energy structure within the group; see Fig. 5d–g for where these landmarks are located on the corresponding energy–density landscapes.

We identified landmark structures for **TH2, TH4, TH5** and **T2** following the same protocol. Since our pore descriptors are agnostic to the molecular structure, landmark structures can be compared across the different molecules in a single projection. For visual comparison, we applied the parametric Uniform Manifold Approximation and Projection (UMAP)[38] technique to learn a mapping from the high-dimensional porosity space to a 2D representation (Fig. 5a), where each point represents a crystal structure and the points are spatially arranged such that the closer the two points are on the plot, the more similar the two structures are in the porosity space. We further used the k-means algorithm[39] to identify clusters on the 2D UMAP space, which are superposed on the 2D UMAP plot (inset, Fig. 5a).

All four experimental polymorphs of **T2**, as well as most of the structures highlighted above for **TH2, TH4** (Fig. 3) and **TH5** (Supplementary Fig. 8), were identified as landmarks on the porosity space; note that **TH2-C, F**, and **TH5-C2, D2** are not shown in Fig. 5a–c because they are not the representative structure (in this case, the most stable structure) of their corresponding cluster. The structures that have isostructural pore channels—for example, **TH2-A, TH4-A, TH5-A** and **T2-γ** all have hexagonal pore channels—are located in close proximity on the 2D UMAP representation (Fig. 5a). An interactive explorer

for the 2D UMAP embeddings of the porosity spaces of **TH2, TH4, TH5** and **T2** is available in our online visualization app (https://www.interactive-esf-maps.app), which allows the user to inspect landmark structures identified by having either the lowest lattice energy or the largest free sphere within the group.

Overall, the structures become more porous, with a higher pore dimensionality and/or a larger accessible surface area, when going from the bottom-right to the top-left (or from k-means group 1 to group 5; see inset in Fig. 5) of the UMAP-embedded porosity space (Supplementary Fig. 11). Most landmark structures exhibit extended hydrogen-bonded networks (Fig. 5b), while some structures also benefit from a complementary stabilization by π–π stacking interactions (Fig. 5c). Results for spiro-linked **SH1, SH2** and **S2** are shown in Supplementary Fig. 12.

ESF maps are simplified representations of complex, high-dimensional structure–property landscapes, providing a powerful visualization of the range of properties and stabilities of the associated crystal structures. However, ESF maps can be challenging to interpret, especially as they become more complex. Analogies with geographical maps break down when the structure–property relationships are encoded by a high-dimensional ESF landscape that may have 10,000 s of structures on a single map. Inspecting ESF maps by eye is laborious and increasingly intractable as the maps become larger, more numerous, and higher-dimensional. The 2D embedding approach shown here makes ESF maps machine readable. To give one use case: it is often desirable to make comparisons between ESF maps for different molecules to assess whether two molecules will be functionally similar or not. This unified embedding approach will be useful for comparing multiple CSP datasets and identifying functionally similar structures using the encoding representation. This might be used, for example, to select the most synthetically accessible molecule in a set of candidates that is likely to express the property of interest, such as a specific pore size. This approach automatically and systematically identifies a small set of landmark structures (typically, 10s to 100s) from the whole CSP landscape (typically, 1000s to 10,000s structures). This allow us to focus more expensive calculations on a smaller set of structures: for example, to carry out solvent stabilization calculations to better assess the synthetic accessibility of specific polymorphs. These calculations are too expensive to perform on entire CSP datasets and more simplistic filtering methods (e.g., using a lattice energy cut-off) may miss key landmark structures.

Simple pore descriptors, such as pore diameters and surface areas, do not have the resolution that is needed to distinguish structures atomistically. By contrast, a range of numerical representations, such as SOAP[40], allow for measuring the similarity between atomistic structures and have been widely used in machine learning tasks[41]. Here, we used SOAP descriptors to encode all the crystal structures of **TH4** and, together with a regularized entropy match (REMatch) kernel[42], to quantify the similarity between every pair of structures. The resulting similarity matrix was then projected onto a 2D space by a UMAP embedding, as shown in Fig. 6a.

For **TH4**, the crystal structures are split, broadly speaking, into two disconnected 'islands' in the SOAP space (Fig. 6a). Both islands contain structures that span the whole density range (Fig. 6b, c). Tracing structures on each island back to the energy–density landscape reveals that the smaller of the two islands (blue dotted square) is overwhelmingly dominated by structures exhibiting 1D pore channels (Fig. 6b), while the larger island (red dotted square) has a greater number of structures with different pore dimensionalities (Fig. 6c). All the landmark structures, **TH4-A** to **F**, are located on the smaller, blue island, as well as structures belonging to the spikes and most of the leading-edge structures on the energy–density landscape (Fig. 6b).

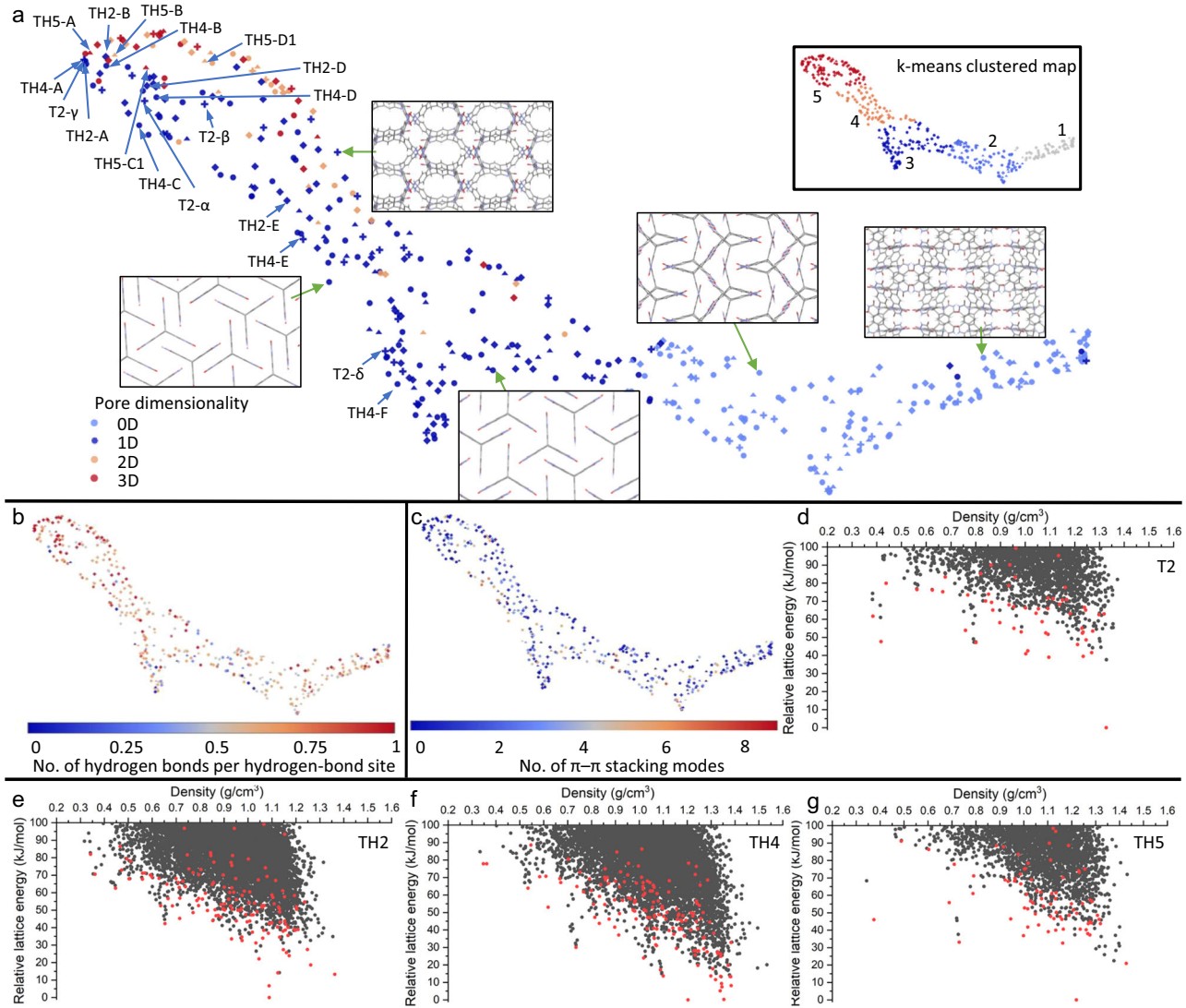

**Fig. 5 Porosity space of the landmark structures of TH2, TH4, TH5 and T2. a–c** 2D UMAP embeddings of the porosity spaces of **TH2** (diamond), **TH4** (circle), **TH5** (triangle) and **T2** (cross), colour coded by the pore dimensionality (**a**), the number (no.) of hydrogen bonds per hydrogen-bond site (**b**), or the total number of π–π stacking modes of the crystal structure (**c**); the symbol size is scaled by the accessible surface area. All the points shown in (**a–c**) are the lowest-energy structures in the respective clusters by affinity propagation and are highlighted in red on their corresponding energy–density landscapes: **T2** (**d**), **TH2** (**e**), **TH4** (**f**) and **TH5** (**g**).

As discussed above, these structures all feature extended hydrogen-bonded chains along the 1D channels. Higher-density structures on the blue island show increased π–π stacking. Almost all structures on the red island are found in the bulk of the energy–structure landscape, featuring diverse packing patterns, which is understandable as it covers a much larger area in the SOAP space than the smaller blue island. For **TH5**, the 2D UMAP embedding of the SOAP space (shown in Supplementary Fig. 15) is not clearly separated into 'islands' but, like **TH4**, the leading-edge structures are mostly located in one region of the embedding.

SOAP descriptors, by design, encode atomic neighbour environments within a cut-off radius, and they are therefore effective at capturing local chemical information such as hydrogen bonding and π–π stacking. A larger cut-off radius of 8.0 Å (Supplementary Fig. 14) results in a similar picture to that found with a cut-off radius of 6.0 Å (Fig. 6). By contrast, SOAP descriptors have been shown to not capture long-range order, such as molecular packing[26], so these projections are complementary to the pore-based descriptor projections shown in Fig. 5.

## Discussion

We have computed ESF maps for a series of molecular tectons that comprise either a triptycene or a spiro-biphenyl core, functionalized with various different hydrogen-bonding moieties, evaluating their abilities to generate porosity in the solid state. Through quantitative analyses of the intermolecular hydrogen bonding and π–π stacking for all the predicted crystal structures, we showed how the ESF maps evolve arising from the different balance of intermolecular interactions in the various tectons. Across the whole series of the molecules studied, intermolecular hydrogen bonding and intermolecular stacking are found to be mostly competing forces in driving the solid-state packing of the molecules. That is, high-porosity, low-density structures are primarily stabilized by extensive hydrogen bonding with minimal intermolecular stacking, while densely packed structures exhibit high levels of stacking but decreased levels of hydrogen bonding. Structures in the intermediate density range are stabilized by a combination of hydrogen bonding and stacking. This results from the positioning of the hydrogen-bonding sites, as well as the number of them, and the contorted molecular core. **TH4** and

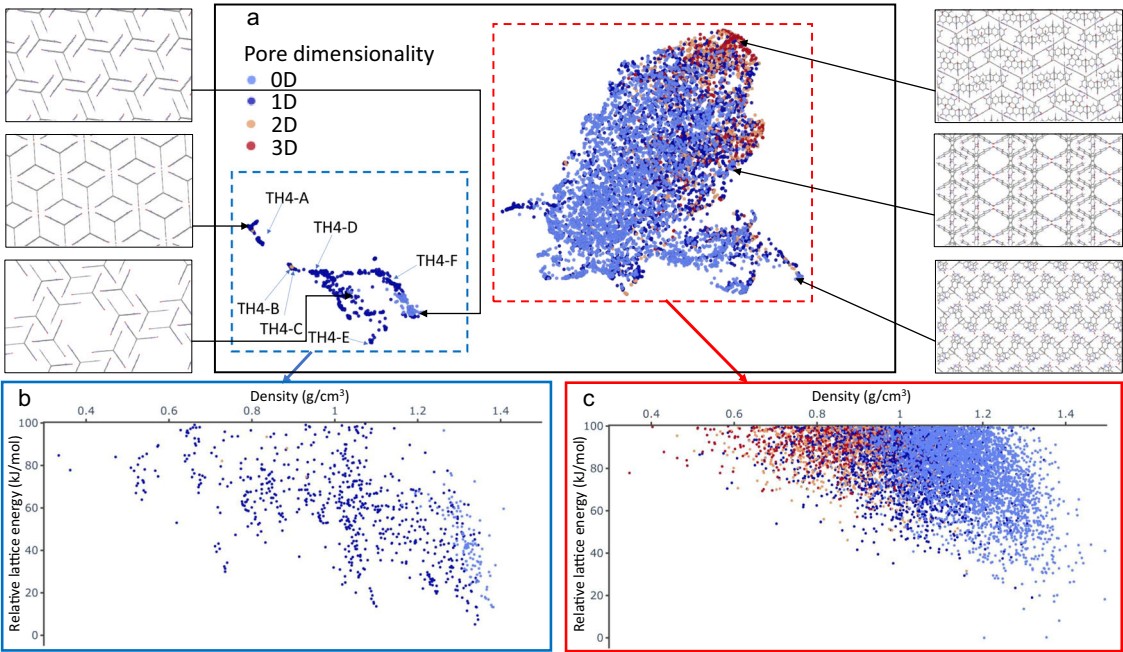

**Fig. 6 The ESF data of TH4 mapped onto its SOAP space. a** 2D UMAP embedding of the SOAP space of **TH4**, colour coded by the pore dimensionality. **b**, **c** Energy–density landscapes correspond to the regions marked out in (**a**), colour coded by the pore dimensionality.

**TH5** have been identified as promising targets for future experimental efforts, because they are both predicted to give multiple (highly) porous crystalline structures that may be experimentally accessible, for example by solvent stabilization. **TH5** has been synthesized before[28], and our results suggest that it would be interesting to re-evaluate this molecule in terms of porosity across a range of crystallization solvents[20].

Inspecting a large and complex multidimensional ESF map can be laborious, even with the help of our interactive ESF Explorer (https://www.interactive-esf-maps.app). Here, we have demonstrated a simple and general framework for representing the high-dimensional data of ESF maps and for systematically identifying 'landmark' structures on the map. By applying unsupervised learning to pore descriptors, as well as SOAP representations, two-dimensional embeddings of the high-dimensional ESF data could be learned, which are human interpretable. Our approach of encoding, learning, and representing ESP maps enables an efficient navigation of the complex ESF space within a unified framework, allowing us to automatically identify energetically favourable or functionally interesting structures across different systems, as well as revealing complex structure–function correlations that are hidden when inspecting individual structural features. This marks a step toward an automated analysis of high-throughput computation of ESF maps, which will be beneficial in facilitating autonomous searches for functional molecular crystals in the future—for example, to create machine-readable maps to prioritize automated robotic searches[43,44].

## Methods

**Crystal structure prediction (CSP).** Geometries of all the molecules studied were fully optimized at the B3LYP/6-311G(d,p) level of theory, using the Gaussian16 software[45], followed by frequency calculations to ensure that they are all true local minima. These molecular geometries were held rigid throughout crystal structure generation and lattice energy minimization.

Trial crystal structures were generated with one molecule in the asymmetric unit for the 23 most common space groups: $P2_1/c$ (34.4%), $P\bar{1}$ (24.8%), $C2/c$ (8.4%), $P2_12_12_1$ (7.1%), $P2_1$ (5.1%), $Pbca$ (3.3%), $Pna2_1$ (1.4%), $Pnma$ (1.1%), $Cc$ (1.0%), $P1$ (1.0%), $C2$ (0.8%), $Pbcn$ (0.8%), $Pca2_1$ (0.7%), $R\bar{3}$ (0.7%), $P2/c$ (0.6%), $C2/m$ (0.5%), $P2_1/m$ (0.5%), $Pc$ (0.4%), $P2_12_12$ (0.4%), $I4_1/a$ (0.4%), $Pccn$ (0.4%), $Fdd2$ (0.3%), and $P4_2$ (<0.3%); the values in the brackets are relative frequencies of the space groups reported in the Cambridge Structural Database.

CSP was performed using a quasi-random sampling procedure, as implemented in the Global Lattice Energy Explorer software[31]. The generation of crystal structures involved a low-discrepancy sampling of all structural variables within each space group: unit cell lengths and angles, and molecular positions and orientations within the asymmetric unit. Space-group symmetry was then applied, and a geometric test was performed for overlap between molecules, which was removed by lattice expansion (the SAT-expand method in ref. [31]). Lattice energy calculations were performed with an anisotropic atom–atom potential using DMACRYS[46]. Electrostatic interactions were modelled using an atomic multipole description of the molecular charge distribution (up to hexadecapole on all atoms) from the B3LYP/6-311G(d,p)-calculated charge density using a distributed multipole analysis[47]. Atom–atom repulsion and dispersion interactions were modelled using a revised Williams intermolecular potential[48], which has been benchmarked against accurate, experimentally determined lattice energies for a range of molecular crystals[49], and was applied successfully in our earlier CSP studies of **T2** and the related imide **T1**, reproducing the known crystal structures[3]. Charge–charge, charge–dipole and dipole–dipole interactions were calculated using Ewald summation; all other intermolecular interactions were summed to a 25-Å cut-off between molecular centres of mass. All accepted trial structures were lattice energy-minimized, and the search was run until a total of 5000 lattice energy minimizations had been performed in each space group.

Removal of duplicate structures was performed in two steps. First, all structures within a lattice energy window of 1.0 kJ mol$^{-1}$ and within a density window of ±0.05 g cm$^{-3}$ were compared using powder X-ray diffraction (PXRD) patterns generated by Platon[50] (wavelength: 0.7 Å; two-theta range: 20°) using a constrained dynamic time-warping method to compare pairs of structures. Structures were considered a match when the Euclidean distance between the PXRD patterns (normalized by area) was <10. This was followed by using the COMPACK[51] algorithm for clustering: 1.0 kJ mol$^{-1}$ and ±0.05 g cm$^{-3}$ selection windows; a distance tolerance of 40% and a maximum value of the RMSD of 0.4 Å for 30 molecules.

**Pore-geometry analysis.** Topological analysis of the pore space within a crystal structure was performed using the void analysis tool zeo++[52]. The outputs from this analysis included the pore dimensionality (0D, 1D, 2D or 3D), pore diameters, surface areas and pore volumes. A probe radius of 1.70 Å was used in all calculations. A total of 18 pore descriptors were used to describe the porosity space of the predicted crystal structures, with full details of their definitions given in Supplementary Methods. These 18 descriptors are simple extensions to four basic pore descriptors: crystal density, largest pore diameter, total surface area and total pore volume. First, the total surface area and the total pore volume were decomposed into accessible and non-accessible contributions. Second, to capture the heterogeneity of the pore geometry within a structure, several descriptors were derived based on the surface areas and pore volumes of individual channels and pockets. We found that this set of descriptors satisfactorily captured different pore shapes, such as those having multiple channels with different pore widths or having both channels and pockets.

**Hydrogen-bond and π–π stacking analysis**. For each predicted crystal structure, hydrogen bonds were identified with the following limits on geometry: $r_{H\cdots A} < $ [sum (van der Waals radii[53] of H and A)] $- 0.1$ in Å and $\angle D-H\cdots A > 100°$, where D and A are the hydrogen-bond donor and acceptor atoms, respectively. Intermolecular stacking was quantified as the number of face-to-face π–π stacking between two molecular arms, which was identified by the distance between the centroids of two neighbouring aromatic rings being less than 4.4 Å and the dihedral angle between the two ring planes being less than 35°. The CSD Python Application Programming Interface, together with in-house scripts, was used to perform these analyses.

**Visualization of the porosity space and the SOAP space**. The UMAP technique was used for dimensionality reduction for mapping high-dimensional data to 2D representations, while preserving both global and local topological structures of the data in the high-dimensional space as much as possible. That is, the points are arranged spatially such that the closer the two points are on the 2D plot, the more similar the two molecules are, as described by the encoding descriptors. For the porosity spaces (Fig. 5a, Supplementary Figs. 11–13), the pairwise distances between crystal structures were computed as the Euclidean distances between vectors of the pore descriptors. For the SOAP spaces (Fig. 6, Supplementary Figs. 14 and 15), SOAP descriptors were generated for all atoms in the crystal structure, using the DScribe package[54]. The regularized entropy match (REMatch)[42] kernel was used to measure global similarity between crystal structures from SOAP-encoded local atomic environments.

## Data availability

All the predicted crystal structures and properties are available at https://doi.org/10.5258/SOTON/D1602. Data for the ESF maps of **TH4**, as well as data for the 2D embedded porosity spaces and SOAP spaces of **TH2**, **TH4**, **TH5** and **T2**, can be visualized online at https://www.interactive-esf-maps.app.

## Code availability

Python scripts to create interactive visualization tools, like the ESF Explorer shown in this study, are available at https://github.com/Yuchees/esf_explorer_templates[55].

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

## Acknowledgements

We acknowledge the Leverhulme Trust via the Leverhulme Research Centre for Functional Materials Design for funding. C.Z. acknowledges the financial support from the China Scholarship Council (No. 201806740038). Y.L. and H.L. acknowledge the financial support from the National Natural Science Foundation of China (Nos. 91834301, 51621002). We thank Dr. Peter R Spackman and Dr. Marc A. Little for useful discussions.

## Author contributions

C.Z. performed the crystal structure predictions, structural analyses and descriptor calculations. C.Z. and Y.C. performed the unsupervised learning tasks. Y.C. developed and deployed the online application for interactive data visualization. Z.P. proposed the potential synthetic routes and advised C.Z. on crystal structure prediction. X.W. contributed to the interpretation of data and to the implementation of the web-based visualization application; Y.L. and H.L. contributed to the discussions. L.C., A.I.C. and G.M.D. conceived the project; L.C. supervised the project. L.C., C.Z. and A.I.C. led the writing of the manuscript with contributions from all co-authors.

## Competing interests

The authors declare no competing interests.
