## [Peer Review File · Nature Communications]

REVIEWERS' COMMENTS

Reviewer #1 (Remarks to the Author):

SUMMARY

In this work, the authors used energy-structure-function (ESF) maps (previously published) and used them to quantify intermolecular hydrogen bonding and stacking in a set of triptycene- and spiro-biphenyl-based crystal structures, showing how the position and # of hydrogen bonding sites influences the shape of the ESF map. The authors conclude that the position of hydrogen bonding groups plays a more significant role in the porosity than the # of hydrogen bonding sites, at least for the structures they studied. They show how ESF maps can be used for navigating promising structure-function spaces, potentially guiding experiments (they even suggested some interesting structures that could be worth exploring experimentally). The authors also developed a very nice online application for visualizing the ESF maps. Finally, the authors used active learning to learn 2D embeddings of ESF data, which can be used to find energetically favorable and/or interesting structures and (in the future) guide automated experiments.

This paper was overall extremely well written; a few suggestions on how to improve the paper are given below, largely minor points. The results support the conclusions, and I believe there is enough detail to reproduce the results. I want to highlight that the introduction is very well organized and provides a good entrance into the rest of the paper, motivates well the advantages of ESF maps (exploring candidate structures, pre-screening, etc), followed by a good summary of current challenges (computational expense of certain property prediction), and then the limitations of the current ESF maps (missing interesting structures in simple 1D representations). The authors propose to address the limitations by learning 2D embeddings of the high-dimensional ESF data. Furthermore, the online application is very nice and looks like it was a lot of work to create. It is great that the authors provide the template for the online application for other researchers to use (although a few suggestions on how to improve the repo are given below).

MAJOR POINTS

--Something that is missing IMO is a bit longer discussion of the advantages of the UMAP embeddings of the porosity and SOAP spaces e.g. are there interesting structures were found that would have been missed otherwise? The authors mention that these maps can be useful for guiding automated experiments, but it is not 100% clear to me that any new, promising, synthetically accessible structures were identified that weren't found from the ESF maps. I see that the landmark structures from the ESF maps are labeled, so then are we to conclude that their nearest neighbors in the porosity/SOAP maps are equally promising (my intuition is no...)?

--In Figures 2c and 2e, why is the maximum number of HBs that TH4 and TH5 can participate in 12? Shouldn't it be 18? Or is it smaller due to packing constraints? (I understand why it would be 12 for T2, but not for TH4 and TH5)

--In Figures 2b, 2d, and 2f, I'm surprised that there are many high density structures for each molecule that have no/few pi-stacking interactions. To me it doesn't make sense that one could get high density structures without pi-stacking. Is there a rationale behind this?

--Did the authors try using fewer and/or a different set of porosity descriptors than what is listed in the paper for the porosity UMAP embeddings? It isn't clear to me exactly why these descriptors were chosen, or if better embeddings could actually be obtained if fewer descriptors were used (since they contain somewhat redundant information). It would be interesting to compare different maps obtained with different descriptors used to represent the structures, to see what is the best representation. Perhaps this has already been done in another publication, and that is why the authors used the representation they did - but if this is the case, it should be more explicit. Finally, were all the pore descriptors computed using Zeo++? (been some years since I used it, didn't remember Zeo++ could calculate all these descriptors).

MINOR POINTS

--Page 3, in the introduction, the sentence starting "Here, we explored the in silico computational design of..." seems like it was supposed to be a new paragraph, since it is a bit disconnected from the first half of the paragraph.

--The first paragraph of Page 4 is a bit confusing to me the way it is currently written. What I have

interpreted this is as 4-pyridone, 2-pyridone, 2,6(1H,3H)-pyridinedione, 2,4(1H,3H)-pyrimidinedione and 1,4(2H,3H)-pyrazinedione were combined with S2 and T2 to form the 7 molecules studied in this work (TH1, TH2, etc). However, I don't understand exactly how these were "combined" - I am guessing just overlapping the ring systems of the core and the hydrogen bonded moieties, but then shouldn't there be then 10 new molecules, or were three of these previously studied? My suggestion is to rephrase this paragraph a bit to make it more clear what was done and how the final molecules were selected.

--In Figure 1, some of the points from the OD-3D-porosity structures are overlapping, and it is unclear to what extent the distributions of the different classes of structures are overlapping (because the points corresponding to 3D-pore structures were plotted last). This mainly applies to Figure 1c. My suggestion is to "shuffle" the order of plotting so that the distribution of densities in the different classes of structures is more clear (i.e. shuffle the plotting order instead of plotting first the gray points, then the red, then the blue, and finally the green). I would suggest a similar thing for Figure 2, as in the text the authors say that most structures "do not show a large number of hydrogen bonds" but from Figure 2 it looks like at least half of the structures have >6 HBs. However, it could simply be that the red/yellow points were plotted last. I understand that this is one of the limitations of the ESF maps as one starts to plot more and more structures on them.

--Page 4, second paragraph, suggest to explicitly say TH1-4 = {TH1, TH2, TH3, and TH4} (or something along these lines). It took me some time to realize that this was not shorthand for new structures, but that the authors were referring to sets of structures. Maybe other readers will be similarly confused (then again, figured it out in the end).

--Page 6, assuming "ref4" was supposed to be ref (superscript) 4?

--Figure 6, there is a description missing for Figure 6b. On a related note, in page 14 the authors refer to the "blue" island and the "red" island, but they are both mostly blue. Maybe it would be clearer to refer to them as the small island and the large island.

--Recommend to add a paragraph to the methods section on the approaches used to learn the 2D UMAP embeddings from the porosity descriptors as well as from the SOAP descriptors. Makes it easier to quickly scan the paper. The description of these methods is currently buried in the results.

--Journal names in the references have mixed case.

--I know I am not reviewing the GitHub repo containing the app templates, but my impression after visiting the repo was that I wouldn't know where to begin (if I wanted to use the templates) as there is not even a README (other than, a file with this name exists). While it is great that some code is being made publicly available, the GitHub repo is not well documented at the moment, and this is going to prevent people from using the tool. I recommend to improve the documentation in the GitHub repo; not only will it increase the chances that someone uses the templates, but also it is just good practice.

--The link to the crystal structures and properties (<https://doi.org/10.5258/SOTON/D1602>) doesn't actually point to anything, wasn't sure if this would be updated upon publication of the paper but if not just wanted to point it out.

--In Figure S1, in the molecular sketch of S2, seems there are 4 double bonds missing, one in each benzimidazolone arm of the spiro-biphenyl core (since benzimidazolone should be aromatic).

RECOMMENDATION

Accept with minor revisions.

Reviewer #2 (Remarks to the Author):

The authors performed a high throughput screening to the end of crystal structure prediction as well as the construction of energy-structure-function maps of hydrogen-bonded organic frameworks. Furthermore, the authors also provide a protocol to reduce the high-dimensional ESF space onto 2D UMAP-based projections that allow for a more clear indication of correlations present in the ESF space. As such, the authors were able to reveal the correlation between porosity (not only degree of porosity but also which kind) and the ability to form hydrogen bonds and pi-pi stacking. These kind of insights are crucial for an application-oriented design of new materials. Finally, all this is implemented in a web application, allowing external users to navigate the ESF space efficiently. This also greatly enhances the reproducibility of the results and even allows external users to possibly extend towards specific ESF maps not included in the current manuscript.

In my opinion, the research is done in a solid way and the results are presented clearly. I think that the various tools and protocols provided are very useful for the community and therefore value this work of high quality. I recommend publication, possibly taking into account a few minor suggestions I will discuss in the remainder of this report.

1) I was missing a concise summary of the used methodology on how the ESF were constructed. It is indeed described in the Methods section, but I would have appreciated a one-sentence summary in the Results section when the ESFs are first introduced (including a clear reference to the Methods section).

2) As the molecular tectons are rigid during the structure optimization, the force field used for energy calculations only requires non-bonded contributions. Here, the authors use an electrostatic contribution based on the distributed multipole analysis applied on the molecular tectons as well as a Williams potential for the repulsion-dispersion contributions. Since the crystal bonding is dominated by hydrogen bonds and pi-pi stacking, which are not trivial to describe accurately with force fields, I wonder to what degree the force field was tested/validated to describe these interactions. More specifically, is there evidence indicating that the results presented in the paper are not too sensitive to the choices made in the force field model? I do not expect a full comparison using various force field models, as that would be unfeasible within reasonable time, but maybe the authors can comment on the choices made and their rationalizations.

3) The provided web application is indeed very useful as well as user friendly. Using the application I was able to reproduce most of the figures reported in the manuscript (at least the figures that related to TH4). However, it seems that it was not able to reproduce the SOAP-based UMAP-projected 2D map (i.e. figure 6 of the manuscript). Is there a specific reason for this, or do the authors intend to implement this feature at a later time? Finally, I would also make a small suggestion with respect to the web application, that is to allow the user to download the structure of a specific point selected on the map (I did not find this feature). I believe this would greatly increase the usefulness of the application for researchers that want to use this work as a starting point for further more in-depth analysis (e.g. perform dynamical simulations) of the candidate structures identified by this work. Now, the user would need to ask the authors for the structure each time they want to proceed with a certain candidate.

Responses to reviewers' comments

Reviewer #1 (Remarks to the Author):

MAJOR POINTS

--Something that is missing IMO is a bit longer discussion of the advantages of the UMAP embeddings of the porosity and SOAP spaces e.g. are there interesting structures were found that would have been missed otherwise? The authors mention that these maps can be useful for guiding automated experiments, but it is not 100% clear to me that any new, promising, synthetically accessible structures were identified that weren't found from the ESF maps. I see that the landmark structures from the ESF maps are labeled, so then are we to conclude that their nearest neighbors in the porosity/SOAP maps are equally promising (my intuition is no...)?

Response: The ESF map of a molecule represents the high-dimensional energetic, structural, and functional landscape of possible crystal structures. In our original study (*Nature*, **2017**, 543, 657), ESF data were mapped onto individual structural descriptors, such as crystal density or pore diameter. For some molecules (e.g., **T2**), obvious spikes emerged from the bulk of the energy–density landscape, hence prompting us to experimentally target structures in the spikes, particularly the lowest-energy structure of each spike (*i.e.*, at the tip of the spike; the structure is often identified as a landmark structure). If we had only used the single ESF map based on the energy–density landscape (Figure 2c of the 2017 *Nature* study), then we could have missed **T2- α** because it was not the most stable structure in its spike. However, **T2- α** was the most stable structure of its spike in the energy– D_f (a pore diameter) landscape (Figure 3e of the 2017 study). In this new work, the landmark structures A–F for **TH2** and **TH4**, as well as A–D for **TH5**, were identified by inspecting the spikes of multiple ESF maps projected onto different structural descriptors (Figures 3 and S8); this is somewhat laborious, but as for the published **T2- α** case, these structures would not be identified from a single ESF map.

The key advantage of the UMAP embedding approach is that all of these intuitively identified landmark structures are automatically highlighted within a unified framework. First, all the predicted crystal structures are encoded by a set of pore descriptors. Then, they are clustered into groups by structural similarity in the porosity space, with each group assigned a representative/landmark structure that is most energetically favourable or most functionally interesting. In Figure 5a, all the datapoints correspond to the most stable structures of their respective groups. They are all interesting, in principle, because they are structurally diverse and are most likely to be experimentally accessible within their own groups.

We agree with the reviewer that predicting synthetic accessibility of molecular crystals goes beyond comparing their crystal lattice energies – not least because of the prevalence of kinetic polymorphs. Hence, it will often be difficult, if not impossible, to pinpoint a few specific structures on the map that are going to dominate in lab experiments. The reviewer is also correct in saying that two structures being in close vicinity on the embedded pore/SOAP map does not translate into equal probabilities for them to be accessed experimentally, because this proximity relates to structural similarity, but the crystal packings may have nonetheless have quite different stabilities. Nonetheless, the use of UMAP embeddings allows us to consider a much

smaller set of landmark structures (typically, a few 10's), rather than the whole CSP landscape (1,000s to 10,000s structures), and this is extremely helpful for guiding experiments. For example, in terms of choosing molecules to investigate in the first place, we can rapidly assess (albeit not fully quantify) the general likelihood of a given molecule expressing a particular property of interest. We note here that the timescale for the CSP predictions is now on the order of days: that is, much faster than the time required to synthesize a new molecule that is then found not to have any useful crystal packings. Once the selection of target molecule has been made, these UMAP embeddings can assist the identification of crystal structures: for example, by allowing us to compare experimental powder X-ray diffraction (PXRD) data with a much smaller, focused set of landmark structures. For the future, the reduction of the number of candidate structures using UMAP embeddings will allow us to perform more sophisticated solvent stabilization calculations (see *e.g.*, Figure 5 of the 2017 study) to assess the effect of crystallization solvent: these calculations are currently much too expensive to be carried out on entire CSP datasets.

To enhance our discussion of the advantages of this new approach, we have added the following text:

“ESF maps are simplified representations of complex, high-dimensional structure-property landscapes, providing a powerful visualization of the range of properties and stabilities of the associated crystal structures. However, ESF maps can be challenging to interpret, especially as they become more complex. Analogies with geographical maps break down when the structure–property relationships are encoded by a high-dimensional ESF landscape that may have 10,000s of structures on a single map. Inspecting ESF maps by eye is laborious and increasingly intractable as the maps become larger, more numerous, and higher-dimensional. The 2D embedding approach shown here makes ESF maps machine readable. To give one use case: it is often desirable to make comparisons between ESF maps for different molecules to assess whether two molecules will be functionally similar or not. This unified embedding approach will be useful for comparing multiple CSP datasets and identifying functionally similar structures using the encoding representation. This might be used, for example, to select the most synthetically accessible molecule in a set of candidates that is likely to express the property of interest, such as a specific pore size. This approach automatically and systematically identifies a small set of landmark structures (typically, 10's to 100's) from the whole CSP landscape (typically, 1,000's to 10,000's structures). This allow us to focus more expensive calculations on a smaller set of structures: for example, to carry out solvent stabilization calculations to better assess the synthetic accessibility of a specific polymorphs. These calculations are too expensive to perform on entire CSP datasets and more simplistic filtering methods (*e.g.*, using a lattice energy cut-off) may miss key landmark structures.”

--In Figures 2c and 2e, why is the maximum number of HBs that TH4 and TH5 can participate in 12? Shouldn't it be 18? Or is it smaller due to packing constraints? (I understand why it would be 12 for T2, but not for TH4 and TH5)

Response: The maximum number of hydrogen bonds that **TH4** and **TH5** can participate in is 12 for the following reasons. First, in both molecules, the number of hydrogen-bond donors (N–H groups) is 6, which is equal to the number of hydrogen-bond acceptors (O atoms). Second, there is only one unique molecule in the unit cell in these predictions (*i.e.*, $Z' = 1$). So, hypothetically, if 3 out of the 6 O atoms of **TH4/TH5** formed two hydrogen bonds each, using all 6 available N–H groups, then the remaining 3 O atoms would be left without any N–H groups to hydrogen bond with; hence, the maximum number of hydrogen bonds formed per molecule is 12. Effectively, the maximum number of hydrogen bonds that we can anticipate for any of the molecules studied here is two times the number of the hydrogen-bond donors in the molecule.

--In Figures 2b, 2d, and 2f, I'm surprised that there are many high density structures for each molecule that have no/few pi-stacking interactions. To me it doesn't make sense that one could get high density structures without pi-stacking. Is there a rationale behind this?

We thank the reviewer for pointing this out – on reflection, it is not very clear in the text what exactly we mean by ‘ π – π stacking’. In this study, we used a narrow definition of π – π stacking when quantifying the number of stacking modes in each structure, which was given in the Methods section as: “*Intermolecular stacking was quantified as the number of **face-to-face** π – π stacking between two **molecular arms**, which was identified by the distance between the centroids of two neighbouring aromatic rings being less than 4.4 Å and the dihedral angle between the two ring planes being less than 35°.*”

This definition excludes numerous conformations of π stacking, such as T-shaped and parallel-displaced stacking modes. Our motivation for using such a narrow definition was to investigate the effects of stacking between two molecular arms on crystal packing. For example, it was found that such stacking modes are a complementary stabilizing force for some of the structures that belong to a spike. We have now emphasized this definition, as well as referring the reader to the Methods section, which makes this point clear.

--Did the authors try using fewer and/or a different set of porosity descriptors than what is listed in the paper for the porosity UMAP embeddings? It isn't clear to me exactly why these descriptors were chosen, or if better embeddings could actually be obtained if fewer descriptors were used (since they contain somewhat redundant information). It would be interesting to compare different maps obtained with different descriptors used to represent the structures, to see what is the best representation. Perhaps this has already been done in another publication, and that is why the authors used the representation they did - but if this is the case, it should be more explicit. Finally, were all the pore descriptors computed using Zeo++? (been some years since I used it, didn't remember Zeo++ could calculate all these descriptors).

Response: Previously, we showed that a set of four most common descriptors for porous materials—crystal density, total surface area, total pore volume, and largest pore size—clustered together porous structures with similar pore size and surface area, but not necessarily having the same pore shape (Figure 6 in *Chem. Sci.*, 2020, 11, 5423). We have now included Figure S13 to show the porosity spaces of **TH2**, **TH4**, **TH5** and **T2**, encoded by these four pore descriptors. Visual inspection of the structures grouped together confirmed that these four descriptors, understandably, are too coarse-grained to have enough resolution to adequately distinguish different pore geometries.

The 18 pore descriptors used here are simple extensions to those four basic descriptors. First, the total surface area and the total pore volume were decomposed into accessible and non-accessible contributions. Second, to capture some extent of the heterogeneity of the pore geometry within a structure, we used several derived quantities based on the surface areas and pore volumes of individual channels and pockets. We found that this set of descriptors satisfactorily captured different pore shapes, such as those having multiple channels with different pore widths or having both channels and pockets. Indeed, using these 18 descriptors, together with a handful more based on further decomposing the surface area into elemental contributions, we were able to machine learn gas adsorption in the predicted crystal structures of **T2** and those of some other molecules (DOI: 10.26434/chemrxiv.13019960.v1). We have now added a short discussion to ‘Pore-geometry analysis’ in the Methods section.

All the pore analyses were performed using Zeo++. Descriptors 1 to 8 (section 1.1, Supplementary Information) were direct outputs from Zeo++, while descriptors 9 to 18 were calculated, using the equations shown, based on the channel- and pocket-specific results, also provided by Zeo++.

MINOR POINTS

--Page 3, in the introduction, the sentence starting "Here, we explored the in silico computational design of..." seems like it was supposed to be a new paragraph, since it is a bit disconnected from the first half of the paragraph.

Response: Yes, it should have been a separate paragraph, which has now been amended.

--The first paragraph of Page 4 is a bit confusing to me the way it is currently written. What I have interpreted this is as 4-pyridone, 2-pyridone, 2,6(1H,3H)-pyridinedione, 2,4(1H,3H)-pyrimidinedione and 1,4(2H,3H)-pyrazinedione were combined with S2 and T2 to form the 7 molecules studied in this work (TH1, TH2, etc). However, I don't understand exactly how these were "combined" - I am guessing just overlapping the ring systems of the core and the hydrogen bonded moities, but then shouldn't there be then 10 new molecules, or were three of these previously studied? My suggestion is to rephrase this paragraph a bit to make it more clear what was done and how the final molecules were selected.

Response: The reviewer is correct in saying that there should be 10 possible molecules when combining 5 different arms with 2 different cores. However, we only considered two of the

five possible spiro-biphenyl-based molecules, primarily because we found in this study and previously that spiro-linked tetrahedral geometries are less effective at generating porosity than their triptycene-based counterparts. We have now improved our wording in the sentences queried here to make the clear.

--In Figure 1, some of the points from the 0D-3D-porosity structures are overlapping, and it is unclear to what extent the distributions of the different classes of structures are overlapping (because the points corresponding to 3D-pore structures were plotted last). This mainly applies to Figure 1c. My suggestion is to "shuffle" the order of plotting so that the distribution of densities in the different classes of structures is more clear (i.e. shuffle the plotting order instead of plotting first the gray points, then the red, then the blue, and finally the green). I would suggest a similar thing for Figure 2, as in the text the authors say that most structures "do not show a large number of hydrogen bonds" but from Figure 2 it looks like at least half of the structures have >6 HBs. However, it could simply be that the red/yellow points were plotted last. I understand that this is one of the limitations of the ESF maps as one starts to plot more and more structures on them.

Response: We thank the reviewer for this suggestion – we have now included a ‘shuffled’ version of Figure 1 as Figure S1 in the Supplementary Information and pointed the reader to it in the caption of Figure 1.

Figure 2, as in its original form, was indeed plotted without imposing any ordering of the points according to their hydrogen-bond or stacking-mode values; they were ordered by structure ID's. Nonetheless, we have now replotted Figure 2 with randomized ordering of datapoints.

In fact, more than half of the structures for each molecule in Figure 2 do indeed have 6 or more hydrogen bonds: that is, 72%, 69% and 56% for **T2**, **TH4** and **TH5**, respectively. The sentence in question could be misunderstood as we meant that most structures do not simultaneously have a large number of hydrogen bonds and a large number of stacked arms: we have modified the wording slightly to address this.

--Page 4, second paragraph, suggest to explicitly say TH1-4 = {TH1, TH2, TH3, and TH4} (or something along these lines). It took me some time to realize that this was not shorthand for new structures, but that the authors were referring to sets of structures. Maybe other readers will be similarly confused (then again, figured it out in the end).

Response: We have now made this explicit in the text.

--Page 6, assuming "ref⁴" was supposed to be ref (superscript) 4?

Response: It was “ref (space) 4”. We will mend this per journal’s requirement.

--Figure 6, there is a description missing for Figure 6b. On a related note, in page 14 the authors refer to the "blue" island and the "red" island, but they are both mostly blue. Maybe it would be clearer to refer to them as the small island and the large island.

Response: We thank the reviewer for pointing this out. Originally, these islands were referred to as a blue or red one per the coloured dotted box surrounding it in Figure 6a. We have now renamed them as the “small” or “big island”, as suggested. We have also made (more) references to Figure 6b,c in the text where appropriate.

--Recommend to add a paragraph to the methods section on the approaches used to learn the 2D UMAP embeddings from the porosity descriptors as well as from the SOAP descriptors. Makes it easier to quickly scan the paper. The description of these methods is currently buried in the results.

Response: We have now added the relevant details under “Visualization of the porosity space and the SOAP space” in the Methods section.

--Journal names in the references have mixed case.

Response: We have fixed this.

--I know I am not reviewing the GitHub repo containing the app templates, but my impression after visiting the repo was that I wouldn't know where to begin (if I wanted to use the templates) as there is not even a README (other than, a file with this name exists). While it is great that some code is being made publicly available, the GitHub repo is not well documented at the moment, and this is going to prevent people from using the tool. I recommend to improve the documentation in the GitHub repo; not only will it increase the chances that someone uses the templates, but also it is just good practice.

Response: We thank the reviewer for raising this, and we accept that this can be improved. We have now expanded the documentation on the GitHub repository and tidied up the codes so that they can be run as a minimal example. Future efforts will be made on preparing examples/tutorials for key individual functionalities of the interactive visualization application so that it will be easier for others to recreate a fully functional application for new datasets supplied by themselves.

--The link to the crystal structures and properties (<https://doi.org/10.5258/SOTON/D1602>) doesn't actually point to anything, wasn't sure if this would be updated upon publication of the paper but if not just wanted to point it out.

Response: All the data had been deposited to the University of Southampton Institutional Repository, under the specified DOI, prior to the initial submission. The usual policy is that the DOI is made active upon acceptance of the manuscript.

--In Figure S1, in the molecular sketch of S2, seems there are 4 double bonds missing, one in each benzimidazolone arm of the spiro-biphenyl core (since benzimidazolone should be aromatic).

Response: We thank the reviewer for pointing out this oversight of ours. We have now corrected the chemical diagram of S2.

RECOMMENDATION

Accept with minor revisions.

Reviewer #2 (Remarks to the Author):

1) I was missing a concise summary of the used methodology on how the ESF were constructed. It is indeed described in the Methods section, but I would have appreciated a one-sentence summary in the Results section when the ESFs are first introduced (including a clear reference to the Methods section).

Response: We have added the following text as the opening sentence of section ‘ESF data mapped onto individual structural descriptors’:

“Energy–structure–function (ESF) maps combine crystal structure prediction (CSP), which determines the stable crystalline arrangements available to a molecule, with predictions of materials properties of interest, using the molecular structure as the only input (see Methods section for details).”

2) As the molecular tectons are rigid during the structure optimization, the force field used for energy calculations only requires non-bonded contributions. Here, the authors use a electrostatic contribution based on the distributed multipole analysis applied on the molecular tectons as well a Williams potential for the repulsion–dispersion contributions. Since the crystal bonding is dominated by hydrogen bonds and pi-pi stacking, which are not trivial to describe accurately with force fields, I wonder to what degree the force field was tested/validated to describe these interactions. More specifically, is there evidence indicating that the results presented in the paper are not too sensitive to the choices made in the force field model? I do not expect a full comparison using various force field models, as that would be unfeasible within reasonable time, but maybe the authors can comment on the choices made and their rationalizations.

Response: We have not tested the sensitivity of the ESF maps to the choice of force field. However, the performance of the revised W99 force field, combined with atomic multipole electrostatics, has been benchmarked against high quality experimentally determined lattice energies of a range of molecular crystals (*Phys. Chem. Chem. Phys.*, **2016**, 18, 15828), and was used successfully in the past for CSP of rigid, hydrogen-bonding molecules, including the CSP of **T2** and the related imide **T1** in our previous work (*Nature*, **2017**, 543, 657). We have added

text and a reference to the paper reporting the benchmarking of force fields to the Methods section.

3) The provided web application is indeed very useful as well as user friendly. Using the application I was able to reproduce most of the figures reported in the manuscript (at least the figures that related to TH4). However, it seems that was not able to reproduce the SOAP-based UMAP-projected 2D map (i.e. figure 6 of the manuscript). Is there a specific reason for this, or do the authors intent to implement this feature at a later time? Finally, I would also make a small suggestion with respect to the web application, that is to allow the user to download the structure of a specific point selected on the map (I did not find this feature). I believe this would greatly increase the usefulness of the application for researchers that want to use this work as a starting point for further more in-depth analysis (e.g. perform dynamical simulations) of the candidate structures identified by this work. Now, the user would need to ask the authors for the structure each time they want to proceed with a certain candidate.

Response: The UMAP-projected 2D map of the SOAP spaces was not deployed onto our web-based application due to the memory limitation of its hosting server, Heroku (<https://www.heroku.com/home>). Heroku runs applications in lightweight, isolated Linux containers, and our container has a RAM of 512 MB. Since there are more than 8,000 crystal structures of TH4, we found that the user experience was very poor when visualizing this map online: response times were long, interactive features were glitchy and sessions crashed when trying to render many structures at the same time (datapoints are closely located to one another on the map, and box-selecting even a small region could have selected a few tens of structures to render their structural visualizations). So, we decided not to include the SOAP map in our online application.

However, if the user chooses to run this application locally (e.g., on a standard office computer with at least 1 GB RAM available for the application), then it should run smoothly for the SOAP map. To help facilitate this, we have made public our codes for generating such interactive visualization applications, which can be run locally, by the user, and viewed in a web browser or deployed online via the mechanism of their choice.

We thank the reviewer for the feature suggestion: we agree that this would be a great feature for enhancing the interactivity of our application. We have started looking into implementing this feature. In the meantime, for the user to obtain structure files for any points on the maps displayed in the online application, the structure ID's can be used to retrieve the corresponding CIF files from our deposited dataset at <https://doi.org/10.5258/SOTON/D1602>, from which all the predicted crystal structures and properties are available. This link will be made active upon publication of the paper.